# The effect of levocarnitine supplementation on dialysis-related hypotension: A systematic review, meta-analysis, and trial sequential analysis

**Api Chewcharat**[1], **Pol Chewcharat**[2], **Weitao Liu**[1], **Jacqueline Cellini**[3], **Elizabeth A. Phipps**[1], **Jill A. Melendez Young**[1], **Sagar U. Nigwekar**[4]*

**1** Department of Medicine, Mount Auburn Hospital, Harvard Medical School, Cambridge, MA, United States of America, **2** Department of Epidemiology, Harvard T.H. Chan School of Public Health, Boston, MA, United States of America, **3** Countway Library, Harvard Medical School, Boston, MA, United States of America, **4** Division of Nephrology, Department of Medicine, Massachusetts General Hospital, Harvard Medical School, Boston, MA, United States of America

* snigwekar@mgh.harvard.edu

**Data Availability Statement:** All relevant data are within the manuscript and its Supporting Information files.

## Abstract

### Background

Dialysis patients have been shown to have low serum carnitine due to poor nutrition, deprivation of endogenous synthesis from kidneys, and removal by hemodialysis. Carnitine deficiency leads to impaired cardiac function and dialysis-related hypotension which are associated with increased mortality. Supplementing with levocarnitine among hemodialysis patients may diminish incidence of intradialytic hypotension. Data on this topic, however, lacks consensus.

### Methods

We conducted electronic searches in PubMed, Embase and Cochrane Central Register of Controlled Trials from January 1960 to 19th November 2021 to identify randomized controlled studies (RCTs), which examined the effects of oral or intravenous levocarnitine (L-carnitine) on dialysis-related hypotension among hemodialysis patients. The secondary outcome was muscle cramps. Study results were pooled and analyzed utilizing the random-effects model. Trial sequential analysis (TSA) was performed to assess the strength of current evidence.

### Results

Eight trials with 224 participants were included in our meta-analysis. Compared to control group, L-carnitine reduced the incidence of dialysis-related hypotension among hemodialysis patients (pooled OR = 0.26, 95% CI [0.10–0.72], p = 0.01, I² = 76.0%). TSA demonstrated that the evidence was sufficient to conclude the finding. Five studies with 147 participants showed a reduction in the incidence of muscle cramps with L-carnitine group (pooled OR = 0.22, 95% CI [0.06–0.81], p = 0.02, I² = 74.7%). However, TSA suggested that further high-quality studies were required. Subgroup analysis on the route of

**Funding:** SUN is supported by the National Institute of Diabetes and Digestive and Kidney Diseases (1U01DK123818). The content is solely the responsibility of the authors and does not necessarily represent the official views of the the National Institutes of Health. The funders had no role in study design, data collection and analysis, decision to publish, or preparation of the manuscript. There was no additional external funding received for this study.

**Competing interests:** The authors have declared that no competing interests exist.

supplementation revealed that only oral but not intravenous L-carnitine significantly reduced dialysis-related hypotension. Regarding dose and duration of L-carnitine supplementation, the dose > 4,200 mg/week and duration of at least 12 weeks appeared to prevent dialysis-related hypotension.

## Conclusion

Supplementing oral L-carnitine for at least three months above 4,200 mg/week helps prevent dialysis-related hypotension. L-carnitine supplementation may ameliorate muscle cramps. Further well-powered studies are required to conclude this benefit.

## Introduction

Carnitine is a quaternary amine that serves as a transporter of long-chain fatty acids into mitochondria. Carnitine is essential for power generation in the cells, particularly skeletal and cardiac muscle cells as these cells heavily rely on fatty acids as the primary sources of energy [1–3]. Primary sources of carnitine are from endogenous synthesis from kidneys and liver along with dietary intake [4]. Patients on chronic hemodialysis are prone to carnitine deficiency due to impaired production from kidneys in addition to poor dietary intake. Moreover, chronic hemodialysis patients tend to lose carnitine across dialysis membrane during hemodialysis [5–7]. Previous studies reported the association between low blood carnitine level and poor cardiac function, cardiac ischemia, cardiomyopathy, muscle cramps, muscle weakness and hypotension [7,8].

Dialysis-related hypotension is defined as a decrease in systolic blood pressure (SBP) by ≥20 mm Hg or a reduction in mean arterial pressure (MAP) by ≥10 mm Hg associated with a clinical event or the need for intervention during hemodialysis or immediately after hemodialysis [9,10]. The prevalence of dialysis-related hypotension was 15–30% among chronic hemodialysis patients [11]. Dialysis-related hypotension is the consequence of an inadequate cardiovascular response to the reduction in blood volume within a short period of time. Some patients may experience nausea, vomiting, muscle cramps, and chest pain [12,13]. Dialysis-related hypotension is also associated with higher morbidity including vascular access thrombosis, mesenteric ischemia, and mortality [12,14].

Previous studies investigated various treatments to help diminish the incidence of dialysis-related hypotension including levocarnitine (L-carnitine) supplementation. However, a previous meta-analysis conducted by Lynch et al. in 2008 [15] suggested no sufficient evidence to conclude that L-carnitine supplementation could prevent dialysis-related hypotension or muscle cramps. Since then, there were novel studies published and suggested potential benefits of L-carnitine on preventing dialysis-related hypotension. We hypothesized that L-carnitine supplementation may help reduce dialysis-related hypotension episodes leading to a decrease in morbidity and mortality among hemodialysis patients.

Therefore, the main aim of this meta-analysis was to investigate the effect of L-carnitine supplementation on preventing dialysis-related hypotension.

## Material and methods

### Data sources and searches

The protocol for this systematic review is registered with PROSPERO (International Prospective Register of Systematic Reviews; no. CRD42021289065). We conducted electronic searches

in PubMed, Embase and Cochrane Central Register of Controlled Trials from January 1960 to November 2021 to identify randomized controlled trials (RCTs), which explored the effects of L-carnitine supplementation on the incidence of dialysis-related hypotension among chronic hemodialysis patients. Manual searches of the reference lists from all relevant original and review articles were also conducted to identify additional eligible studies. This study was conducted based upon the Preferred Reporting Items for Systematic Reviews and Meta-Analysis (PRISMA) statement [16] provided in **S1 File**. Search strategy was provided in **S2 File**.

### Selection criteria

RCTs examining the effect of L-carnitine supplementation compared to control on dialysis-related hypotension were included. Conference abstracts were excluded. There were no restrictions on age, sample size or study duration. Retrieved articles were individually reviewed for eligibility by two investigators (A.C. and P.C.). If there were any disagreements that did not have a conclusion, a third author (W.L.) would make a consensus.

### Data extraction and quality assessment

The following data were extracted from the included RCTs: authors, year of publication, country of origin, sample size, duration of follow-up, route of L-carnitine supplementation, dose, mean age and proportion of male participants. The following outcomes of interest were examined: incidence of dialysis-related hypotension (primary outcome) and muscle cramps (secondary outcome).

Revised Cochrane risk-of-bias tool for randomized trials (RoB 2) [17] was utilized to assess the risk of bias for RCTs. The assessment included the following components: risk of bias arising from randomization process, risk of bias due to deviation from the intended interventions, missing outcome data, risk of bias in the measurement of the outcome and risk of bias in the selection of the reported result. A judgment regarding the risk of bias arising from each domain is generated by an algorithm, based on answers to the signaling questions which includes high risk of bias, low risk of bias, or some concerns.

### Data synthesis and statistical analysis

Outcomes including odds ratio of dialysis-related hypotension and muscle cramps were summarized by the generic inverse variance approach of DerSimonian and Laird, which designated the weight of each study based on its variance [18]. Random effects models were used owing to the expected clinical heterogeneity in the included populations. All pooled estimates were shown with 95% confidence intervals (CI). We also performed subgroup analysis on the route of L-carnitine supplementation, dose, and duration of treatment. Sensitivity analysis was done as a dialysis technique prior to 1990 was significantly different from the current technique including the use of nonvolumetric ultrafiltration or bioincompatible membranes. Heterogeneity among effect sizes estimated by individual studies was described with the $I^2$ index and the chi-square test. A value of $I^2$ of 0%-25% represents insignificant heterogeneity, 26%-50% low heterogeneity, 51%-75% moderate heterogeneity and 76–100% high heterogeneity [19].

Trial sequential analysis (TSA) for dialysis-related hypotension and muscle cramps was performed to reduce the risk of type I errors from repetitive statistical testing and diminish random type I and type II errors from including small trials and sparse data [20,21]. Furthermore, TSA was utilized to help estimate the sample size for further similar trials required to have adequate power. Trial sequential monitoring boundaries and required sample size were quantified by setting type I error at 5% and power of 80% along with relative risk reduction of 60% for

dialysis-related hypotension and 57% for muscle cramps based upon results after excluding high-risk trials. Heterogeneity between trials was accommodated by applying random effect models.

Publication bias was evaluated using funnel plots and the Egger test to assess for asymmetry of the funnel plot. A p-value less than 0.05 indicates the presence of publication bias [22]. The meta-analysis was performed by STATA/IC 14.1 (StataCorp LLC, Texas, USA) and R studio version 1.4.1103.

# Results and discussion

## Characteristics and quality of the studies

After excluding 80 duplications, a total of 172 potentially relevant citations were identified and screened. Seventeen trials were evaluated in detail, of which 8 trials with 224 participants fulfilled the eligibility criteria and were included in this meta-analysis. The literature retrieval, review, and selection process are displayed in Fig 1.

Characteristics of the individual trials are shown in Table 1. The trials varied in sample size from 9 to 82 patients. From 8 trials, three followed a cross-over design [23–25]. There were 3 trials conducted in North America [24,26,27], 2 trials conducted in Europe [23,28], and 3 trials conducted in Asia [25,29,30]. The mean age of patients ranged from 43.8 to 66.9 years old. The duration of follow-up spanned from 6 weeks to 24 weeks. The dose of L-carnitine supplementation ranged from 3,500 to 6,300 mg/week. After meta-analysis by Lynch et al. [15] was published in 2008, there were 3 more trials published with similar baseline characteristics to previous studies. However, two trials [27,30] supplemented L-carnitine at a higher dose than previous trials prior to the study by Lynch et al.

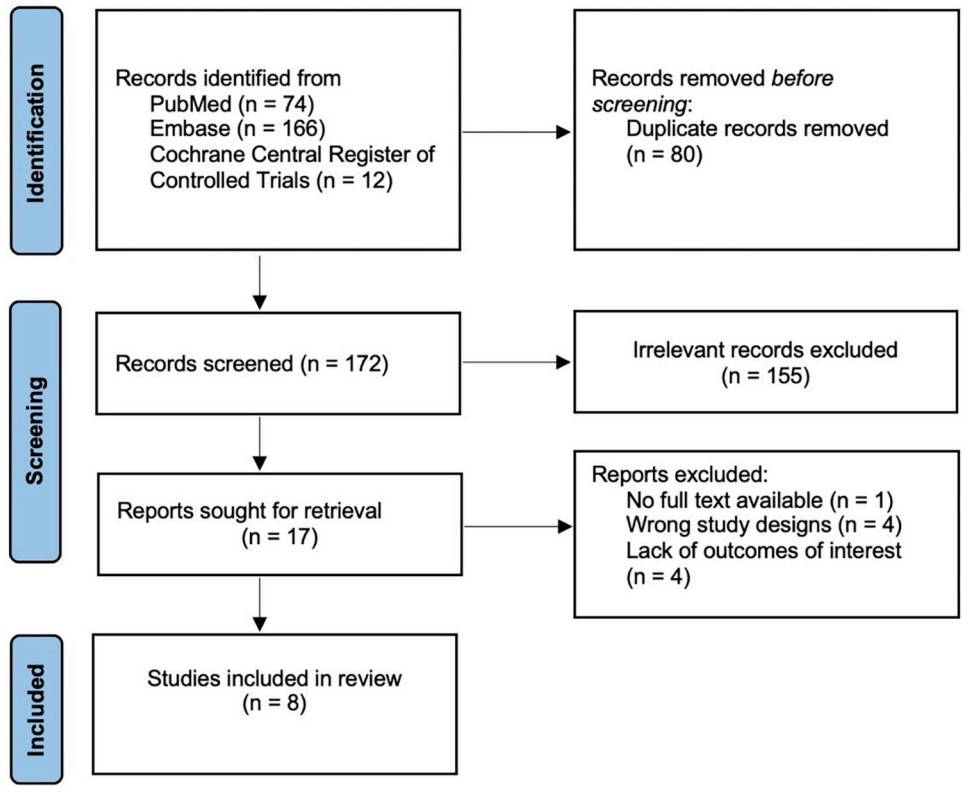

**Fig 1. Search methodology and selection process.**

**Table 1. Main characteristics of studies included in the meta-analysis of the effects of L-carnitine on dialysis-related hypotension.**

| Author | Year of publication | Trial design | Country | Number of patients | Mean age | % Male | Duration of treatment (weeks) | Route of supplementation | Dose of L-carnitine (mg/week) |
|---|---|---|---|---|---|---|---|---|---|
| Casciani et al. | 1983 | Crossover | Italy | 9 | NA | 61 | 8 | Oral | 4620 |
| Ahmad et al. | 1990 | Parallel | USA | 82 | 47.8±2.4 | 62 | 24 | IV | 4200 |
| Semeniuk et al. | 2000 | Crossover | Canada | 16 | 66.9 ±15.9 | 31 | 12 | IV | 4200 |
| Vaux et al. | 2003 | Parallel | UK | 26 | 61.3 ±17.7 | 73 | 16 | IV | 4200 |
| Rathod et al. | 2006 | Parallel | India | 20 | 43.8 ±12.8 | 90 | 8 | IV | 4200 |
| Kudoh et al. | 2013 | Parallel | Japan | 18 | 66.7±7.7 | 44 | 12 | Oral | 6300 |
| Khosroshahi et al. | 2013 | Crossover | Iran | 20 | 51.35 ±12.90 | 55 | 6 | Oral | 3500 |
| Ibarra-Sifuentes et al. | 2017 | Parallel | Mexico | 33 | 46.8 ±14.8 | 55 | 12 | IV | 6300 |

USA, United States of America; UK, United Kingdom; NA, not applicable; IV, intravenous.

### Risk of bias

According to the revised Cochrane risk-of-bias tool for randomized trials, with respect to the overall risk of bias, two studies had low risk of bias [28,30]; fours studies with some concerns for risk of bias [24,26,27,29] and another two studies had high risk of bias [23,25]. In terms of risk of bias arising from the randomization process, two studies had high risk of bias [23,25] and two studies raised some concerns [26,29]. For risk of bias due to deviations from the intended interventions, four studies raised some concerns [23,25–27]. Two studies raised some concerns for missing outcome data. All studies had low risk of bias in the measurement of the outcome and risk of bias in selection of the reported result. There was no study that had high risk of bias in all domains (Table 2).

### Effect of L-carnitine on dialysis-related hypotension

Eight study arms with 224 participants demonstrated a significantly lower odds of dialysis-related hypotension among L-carnitine supplementation group compared to placebo (pooled OR = 0.26, 95% CI [0.10–0.72], p = 0.01, $I^2$ = 76.0%) (Fig 2).

**Table 2. Risk of bias according to revised Cochrane risk-of-bias tool for randomized trials.**

| | Risk of bias arising from the randomization process | Risk of bias due to deviations from the intended interventions | Missing outcome data | Risk of bias in measurement of the outcome | Risk of bias in selection of the reported result | Overall risk of bias |
|---|---|---|---|---|---|---|
| Casciani et al. [23] | High | Some concerns | Low | Low | Low | High |
| Ahmad et al.[26] | Some concerns | Some concerns | Low | Low | Low | Some concerns |
| Semeniuk et al.[24] | Low | Low | Some concerns | Low | Low | Some concerns |
| Vaux et al.[28] | Low | Low | Low | Low | Low | Low |
| Rathod et al.[29] | Some concerns | Low | Low | Low | Low | Some concerns |
| Kudoh et al.[30] | Low | Low | Low | Low | Low | Low |
| Khosroshahi et al. [25] | High | Some concerns | Low | Low | Low | High |
| Ibarra-Sifuentes et al.[27] | Low | Some concerns | Some concerns | Low | Low | Some concerns |

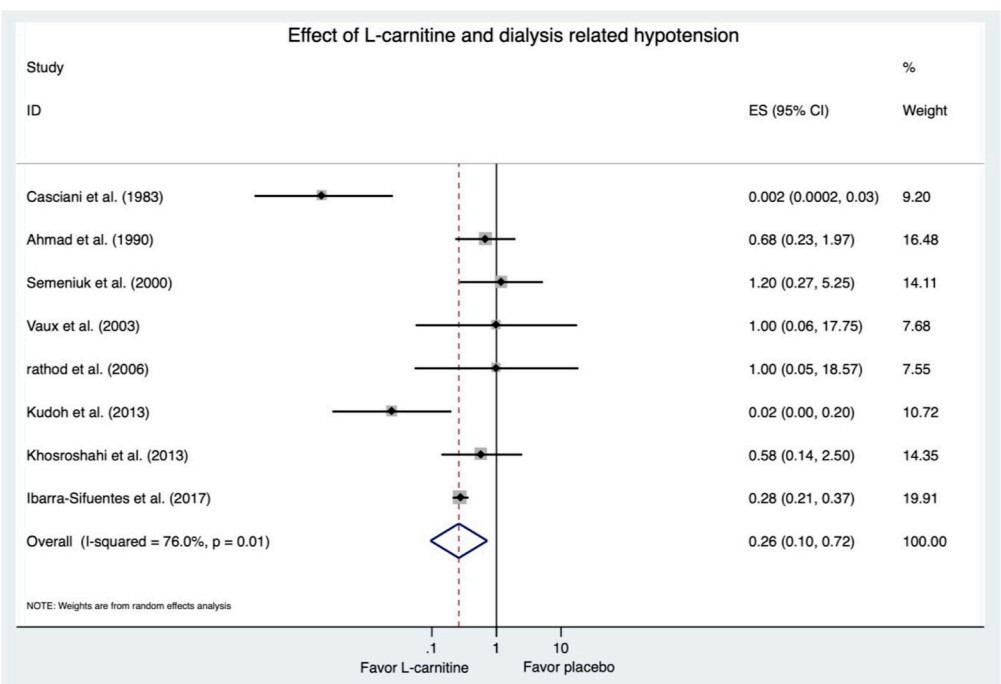

**Fig 2. Forest plots of the included studies assessing odds ratio of dialysis-related hypotension.**

TSA was performed with α of 5% (two-sided), β of 20% and a relative risk reduction of 60% calculated using the incidence rate of trials with low risk of bias. The required information size was estimated to be 172. As shown in Fig 3A, the cumulative Z curve crossed both the required information size and conventional boundary. Thus, given existing evidence, the lower odds of dialysis-related hypotension among hemodialysis patients supplementing with L-carnitine could be considered conclusive.

In the subgroup analysis on route of L-carnitnine supplementation, three studies supplementing L-carnitine orally (47 participants) demonstrated a significant lower odds of dialysis-related hypotension comparing L-carnitine to placebo (pooled OR = 0.03, 95% CI [0.001–0.96], p = 0.04, $I^2$ = 88.3%) while 5 studies supplementing L-carnitine intravenously (177 participants) failed to reveal a difference in incidence of dialysis-related hypotension between these two groups (pooled OR = 0.51, 95% CI [0.25–1.06], p = 0.07, $I^2$ = 42.7%) (Table 3).

Subgroup analysis on dose of L-carnitine supplementation was investigated. Our study used 4,200 mg/week as a cut point as this was a median value. Three studies supplementing L-carnitine > 4,200 mg/week (60 participants) demonstrated a significant lower odds of dialysis-related hypotension comparing L-carnitine group to control group (pooled OR = 0.03, 95% CI [0.001–0.58], p = 0.02, $I^2$ = 90.0%). Five studies supplementing ≤ 4,200 mg/week (164 participants) showed a non-significant difference in dialysis-related hypotension between treatment and placebo group (pooled OR = 0.78, 95% CI [0.39–1.57], p = 0.48, $I^2$ = 0%) (Table 3).

Stratified by the duration of treatment, we utilized 12 weeks as a cut point since this value was a median. Five studies with a duration of treatment at least 12 weeks (175 participants) showed a lower odds of dialysis-related hypotension comparing L-carnitine to control group (pooled OR = 0.37, 95% CI [0.14–0.98], p = 0.04, $I^2$ = 66.4%) while three studies with a duration of treatment less than 12 weeks (49 participants) failed to demonstrate a difference in incidence of dialysis-related hypotension between these two groups (pooled OR = 0.11, 95% CI [0.003–4.40], p = 0.24, $I^2$ = 88.0%) (Table 3).

**a** Trial sequential analysis on the effect of L-carnitine on dialysis-related hypotension

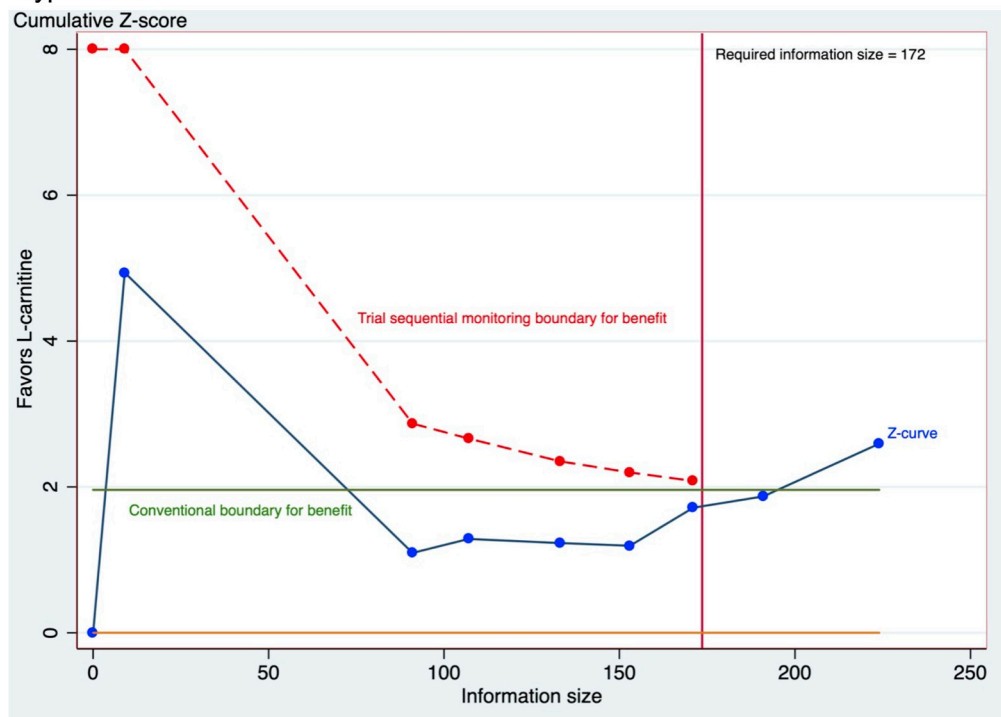

**b** Trial sequential analysis on the effect of L-carnitine on muscle cramps

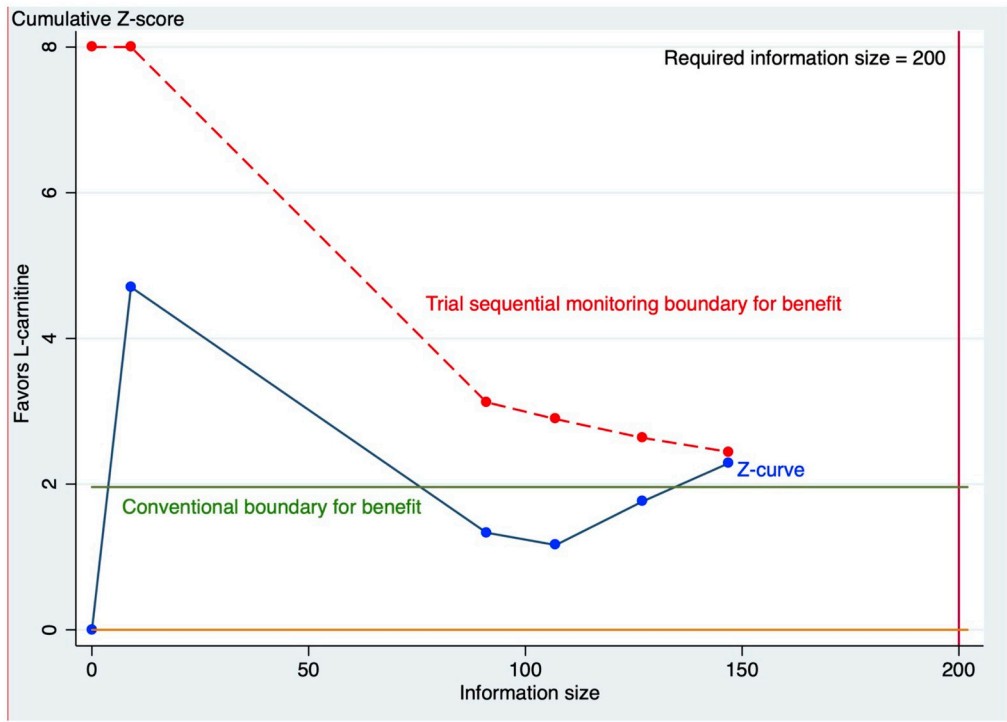

**Fig 3.** a. Trial sequential analysis on the effect of L-carnitine on dialysis-related hypotension. b. Trial sequential analysis on the effect of L-carnitine on muscle cramps.

**Table 3. Summary effects of L-carnitine and subgroup analysis on the route of L-carnitine supplementation, dose and duration of treatment on outcomes of interest among hemodialysis patients.**

| Parameters | Number of studies | Sample Size | OR | 95% CI | P-value | $I^2$ |
|---|---|---|---|---|---|---|
| **Dialysis-related hypotension** | 8 | 224 | 0.26 | (0.10–0.72) | 0.01 | 76.0 |
| Route | | | | | | |
| Oral | 3 | 47 | 0.03 | (0.001–0.96) | 0.04 | 88.3 |
| IV | 5 | 177 | 0.51 | (0.25–1.06) | 0.07 | 42.7 |
| Dose | | | | | | |
| > 4,200 mg/week | 3 | 60 | 0.03 | (0.001–0.58) | 0.02 | 90 |
| ≤ 4,200 mg/week | 5 | 164 | 0.78 | (0.39–1.57) | 0.48 | 0 |
| Duration | | | | | | |
| ≥ 12 weeks | 5 | 49 | 0.37 | (0.14–0.98) | 0.04 | 66.4 |
| < 12 weeks | 3 | 175 | 0.11 | (0.003–4.40) | 0.24 | 88 |
| **Muscle cramps** | 5 | 147 | 0.22 | (0.06–0.81) | 0.02 | 74.7 |

### Effect of L-carnitine on muscle cramps

Five study arms with 147 participants reported a lower odds of muscle cramps comparing L-carnitine group to control group (pooled OR = 0.22, 95% CI [0.06–0.81], p = 0.02, $I^2$ = 74.7%) (Table 3).

TSA was performed with α of 5% (two-sided), β of 20% and a relative risk reduction of 57% calculated using the incidence rate of trials with low risk bias. The required information size was calculated to be 200. As shown in Fig 3B, the cumulative Z curve crossed the conventional boundary but did not cross the trial sequential monitoring boundary indicating that the effect might be false positive. Therefore, further studies are required.

### Sensitivity analysis

Dialysis technique prior to 1990 was significantly different from the current technique including but not limited to the use of nonvolumetric ultrafiltration or bioincompatible membranes. Therefore, we excluded Casciani et al. which was conducted prior to 1990. The pooled odds ratio of dialysis-related hypotension was 0.42 (95% CI [0.20–0.90], p = 0.03, $I^2$ = 54.9%) comparing L-carnitine to placebo. However, regarding muscle cramps, there was no significant difference between the treatment and placebo group (pooled OR = 0.39, 95% CI [0.09–1.07], p = 0.07, $I^2$ = 47.5%).

Subgroup analysis on the design of randomized controlled trials, 3 studies with cross-over trials demonstrated the odds ratio of dialysis-related hypotension was 0.13 (95% CI [0.006–3.10], p = 0.21, $I^2$ = 90%) comparing L-carnitine to placebo. Five studies with parallel trials showed the odds ratio of dialysis-related hypotension was 0.32 (95% CI [0.12–0.82], p = 0.02, $I^2$ = 56.3%) comparing L-carnitine to placebo. With regards to muscle cramps, there were limited number of studies to perform subgroup analysis based on the design of RCTs.

### Assessment of publication bias

Given the number of included studies less than 10, Egger's test and funnel plot were not performed as we do not have adequate power to distinguish chance from real asymmetry [31].

### Discussion

Our study is the largest meta-analysis to assess the treatment effect of L-carnitine on dialysis-related hypotension and muscle cramps. Our study demonstrated the benefits of L-carnitine

**Table 4. Comparison between previous meta-analysis and the present study.**

|  | Lynch et al. (2008) | Our meta-analysis |
|---|---|---|
| Population | Adult patients with end-stage renal disease receiving long-term hemodialysis | Adult patients with end-stage renal disease receiving long-term hemodialysis |
| Data sources | PubMed, Medline, ISI, Ovid, Manual search | Medline, Embase, Cochrane Library (Through Nov 2021) |
| Inclusion | Studies reported either dialysis-related hypotension or muscle cramps | Studies reported dialysis-related hypotension |
| Intervention | L-carnitine | L-carnitine |
| Comparator | Placebo | Placebo |
| No. of RCTs (sample size and study designs) | 4 RCTs with 145 patients (Dialysis-related hypotension) 6 RCTs with 149 patients (Muscle cramps) | 8 RCTs with 224 patients |
| Analytical approach | Random effect model | Random effect model Trial sequential analysis |
| **Results** |  |  |
| Dialysis-related hypotension | Pooled OR = 0.28, 95% CI [0.04,2.23]; p = 0.2, $I^2$ = 80.6% | Pooled OR = 0.26, 95% CI [0.10–0.72], p = 0.01, $I^2$ = 76.0% |
| Muscle cramps | Pooled OR = 0.30, 95% CI [0.09 to 1.00]; p = 0.05, $I^2$ = 70.2% | Pooled OR = 0.22, 95% CI [0.06–0.81], p = 0.02, $I^2$ = 74.7% |
| Subgroup | None | Route, duration of therapy, dose of intervention |

supplementation among hemodialysis patients on preventing dialysis-related hypotension and muscle cramps. However, TSA suggested additional high-quality RCTs are required to confirm the benefit of L-carnitine supplementation on preventing muscle cramps. Subgroup analysis in terms of route, dose, and duration of treatment was performed to obtain more insights on the exploration of heterogeneity. We found that oral L-carnitine supplementation of more than 4,200 mg/week for at least 12 weeks appeared to help diminish dialysis-related hypotension.

A previous meta-analysis by Lynch et al.[15] including 4 RCTs (145 participants) reporting dialysis-related hypotension and 6 RCTs (149 participants) reporting muscle cramps failed to demonstrate benefits for preventing either dialysis-related hypotension or muscle cramps among hemodialysis patients. The nonsignificant findings were likely due to the small sample size. Our meta-analysis included 8 RCTs with 224 participants reporting incidence of dialysis-related hypotension. We found sufficient evidence to support the benefit of L-carnitine in preventing dialysis-related hypotension. Given more studies, we were able to perform subgroup analysis on route, dose, and duration of L-carnitine. The comparative detail of the previous meta-analysis and the present study was illustrated in Table 4.

Carnitine plays a pivotal role in energy metabolism as it transports long-chain fatty acids across the inner mitochondrial membrane and regulates β-oxidation of fatty acids [2]. Hemodialysis patients are commonly found to have carnitine deficiency due to decreased dietary intake, impaired endogenous synthesis, and significant removal by the dialyzer. Carnitine deficiency results in cellular metabolism aberration and impaired energy production because of diminished mitochondrial β-oxidation of fatty acids and accumulation of toxic acylcarnitine suppressing the metabolic activity of various enzymes [2,32]. These impairments and aberrations lead to muscle cramps, dialysis-related hypotension, myopathy, cardiomyopathy, and erythropoietin-resistant anemia among hemodialysis patients [33–35].

Previous studies reported the beneficial effect of L-carnitine supplementation on improving left ventricular mass index (LVMI) and left ventricular ejection fraction (LVEF), particularly among those with left ventricular hypertrophy [36,37] which might help diminish dialysis-related hypotension episodes. Although the exact mechanism remains unknown,

supplementing L-carnitine may help facilitate β-oxidation of fatty acids for cardiac myocytes. This may protect cardiac myocytes from local ischemia and oxidative stress preventing left ventricular end-diastolic dysfunction and cardiac stunning [38–40]. The improvement in cardiac function leads to an increase in cardiac output to counteract the sudden decline in systolic blood pressure during hemodialysis reducing episodes of dialysis-related hypotension [41].

Muscle cramps during hemodialysis are thought to be from hypoxia caused by dialysis-related hypotension, osmotic shifts, and electrolyte disturbances compounded by carnitine deficiency [42]. L-carnitine supplementation may prevent muscle cramps by ameliorating insufficient cellular energy supplies in skeletal muscles via enhancing the rate of β-oxidation of fatty acids, removing short and medium-chain fatty acids, and maintaining glycogen storage [43]. Furthermore, L-carnitine supplementation may help decrease the risk of dialysis-related hypotension contributing to muscle cramps.

L-carnitine can be supplemented orally or intravenously. Our study demonstrated only the benefit in preventing dialysis-related hypotension for oral L-carnitine supplementation. It is possible that doses and frequency of L-carnitine supplementation affect the efficacy of diminishing dialysis-related hypotension. Based on studies included in our meta-analysis, the dose per week of oral L-carnitine was higher than intravenous L-carnitine and the intravenous L-carnitine was supplemented only during hemodialysis sessions 3 times per week. Our study showed that only the dose above 4,200 mg/week was effective for preventing dialysis-related hypotension. Therefore, the significant finding on oral L-carnitine supplementation decreasing dialysis-related hypotension may be driven by the higher dose rather than the route of supplementation per se. Interestingly, orally supplemented L-carnitine is not generally recommended because of limited bioavailability [44] and accumulation of toxic metabolites known as trimethylamine (TMA) and trimethylamine-N-oxide (TMAO) which are associated with cardiovascular events and mortality [45,46]. Given this, intravenous L-carnitine supplementation above 4,200 mg/week may be better to prevent dialysis-related hypotension although is possible that intravenous L-carnitine may be also converted to TMAO as it undergoes enterohepatic circulation [47].

Several strengths are worth mentioning in our study. Firstly, we included only RCTs to limit significant bias and confounding compared to observational studies. Secondly, our study performed the subgroup analysis to examine the effect of L-carnitine supplementation in a specific route, dose, and duration of supplementation on dialysis-related hypotension among chronic hemodialysis patients. Finally, we utilized TSA technique to calculate the required information size and help confirm the reliability of the results. Nevertheless, our study has limitations. First, even though our study had the largest sample size, 224 participants are considered a relatively small number of participants. Second, we acknowledge the heterogeneity in our study was high even after performing subgroup analysis and utilizing a random effect model. The residual confounders such as medications, cardiac function, and different definitions in each study for dialysis-related hypotension and muscle cramps may introduce heterogeneity. Furthermore, subgroup analysis on route, dose, duration of treatment and study design of RCTs could not be performed on muscle cramps due to a limited number of studies. Moreover, due to limited numbers of studies less than 10, methods to investigate for publication bias are underpowered and can be misleading. Therefore, publication bias may occur in our study.

## Conclusions

In summary, the present meta-analysis of 8 RCTs encompassing 224 participants demonstrated that oral L-carnitine supplementation at least 12 weeks with a dose above 4,200 mg/

week could prevent dialysis-related hypotension. Furthermore, L-carnitine supplementation appears to ameliorate muscle cramps, but further high-quality and well-powered studies are required to conclude this beneficial effect.

## Supporting information

**S1 File. Prisma checklist.**
(PDF)

**S2 File. Search strategy.**
(DOCX)

**S3 File. Data abstraction.**
(DTA)

## Author Contributions

**Conceptualization:** Api Chewcharat, Elizabeth A. Phipps, Jill A. Melendez Young, Sagar U. Nigwekar.

**Data curation:** Api Chewcharat, Pol Chewcharat, Weitao Liu, Jacqueline Cellini.

**Formal analysis:** Api Chewcharat, Pol Chewcharat.

**Investigation:** Pol Chewcharat, Jacqueline Cellini, Elizabeth A. Phipps, Sagar U. Nigwekar.

**Methodology:** Api Chewcharat, Jacqueline Cellini, Elizabeth A. Phipps, Sagar U. Nigwekar.

**Supervision:** Elizabeth A. Phipps, Jill A. Melendez Young, Sagar U. Nigwekar.

**Validation:** Jacqueline Cellini, Elizabeth A. Phipps, Sagar U. Nigwekar.

**Visualization:** Pol Chewcharat, Sagar U. Nigwekar.

**Writing – original draft:** Api Chewcharat, Pol Chewcharat, Weitao Liu, Sagar U. Nigwekar.

**Writing – review & editing:** Elizabeth A. Phipps, Jill A. Melendez Young, Sagar U. Nigwekar.

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
