## [Decision Letter · Decision Letter 0]

11 May 2022

PONE-D-22-07106The Effect of Levocarnitine Supplementation on Dialysis-related Hypotension: a Systematic Review, Meta-analysis, and Trial Sequential AnalysisPLOS ONE

Dear Dr. Chewcharat,

Thank you for submitting your manuscript to PLOS ONE. After careful consideration, we feel that it has merit but does not fully meet PLOS ONE’s publication criteria as it currently stands. Therefore, we invite you to submit a revised version of the manuscript that addresses the points raised during the review process.

Please provide a point-by-point response to the comments of the two reviewers.

A marked-up copy of your manuscript that highlights changes made to the original version. You should upload this as a separate file labeled 'Revised Manuscript with Track Changes'.An unmarked version of your revised paper without tracked changes. You should upload this as a separate file labeled 'Manuscript'.If applicable, we recommend that you deposit your laboratory protocols in protocols.io to enhance the reproducibility of your results. Protocols.io assigns your protocol its own identifier (DOI) so that it can be cited independently in the future. For instructions see: https://journals.plos.org/plosone/s/submission-guidelines#loc-laboratory-protocols. Additionally, PLOS ONE offers an option for publishing peer-reviewed Lab Protocol articles, which describe protocols hosted on protocols.io. Read more information on sharing protocols at https://plos.org/protocols?utm_medium=editorial-email&utm_source=authorletters&utm_campaign=protocols.

We look forward to receiving your revised manuscript.

Kind regards,

Mabel Aoun, MD, MPH

Academic Editor

PLOS ONE

Journal Requirements:

(SUN is supported by the National Institute of Diabetes and Digestive and Kidney Diseases (1U01DK123818). The content is solely the responsibility of the authors and does not necessarily represent the official views of the the National Institutes of Health. The funders had no role in study design, data collection and analysis, decision to publish, or preparation of the manuscript.)

3. In your Data Availability statement, you have not specified where the minimal data set underlying the results described in your manuscript can be found. PLOS defines a study's minim4. al data set as the underlying data used to reach the conclusions drawn in the manuscript and any additional data required to replicate the reported study findings in their entirety. All PLOS journals require that the minimal data set be made fully available. For more information about our data policy, please see http://journals.plos.org/plosone/s/data-availability.

Reviewers' comments:

Reviewer's Responses to Questions

**Comments to the Author**

1. Is the manuscript technically sound, and do the data support the conclusions?

Reviewer #1: Yes

Reviewer #2: Yes

2. Has the statistical analysis been performed appropriately and rigorously? 

Reviewer #1: Yes

Reviewer #2: Yes

3. Have the authors made all data underlying the findings in their manuscript fully available?

Reviewer #1: No

Reviewer #2: Yes

4. Is the manuscript presented in an intelligible fashion and written in standard English?

Reviewer #1: Yes

Reviewer #2: Yes

5. Review Comments to the Author

Reviewer #1: In the present systematic review and meta-analysis, the authors investigated the treatment effect of L-carnitine on dialysis-related hypotension and muscle cramps in chronic hemodialysis patients. They included 8 randomized controlled trials with 224 participants, where the mean age of patients ranged from 43.8 to 66.9 years and the duration of follow-up spanned from 6 weeks to 24 weeks. They found a significantly lower odds of dialysis-related hypotension among L-carnitine supplementation group compared to placebo (pooled OR = 0.26, 95% CI [0.10-0.72], p = 0.01, I² = 76.0%), though with high heterogeneity. Further high-quality studies are needed to conclusively prove the effect of L-carnitine on muscle cramps.

The paper is well written, the methodological approach is appropriate and follows PRISMA guidelines, the results are adequately commented upon, and limitations have been correctly examined. The authors have further performed a trial sequential analysis, which helped to conclude the effect of L-carnitine on dialysis-related hypotension. Moreover, subgroup analyses by dose, route and duration of treatment further revealed that oral L-carnitine supplementation at least 12 weeks with a dose above 4,200 mg/week could prevent dialysis-related hypotension.

I have a few minor comments about the manuscript, which I have presented below:

1. Page 5: The authors have stated that “All relevant data are within the manuscript and its Supporting Information files.” Thus, the authors should please amend the data availability statement on Page 4 “No - some restrictions will apply.” PLOS ONE requires all data underlying the findings described fully available.

2. Page 12: Though the authors have found that the lower odds of dialysis-related hypotension among L-carnitine supplementation group compared to placebo was significant, there was high heterogeneity. The authors should please mention this here.

3. Page 13: In subgroup analysis by dose of L-carnitine supplementation, the following statement requires correction from “Three studies supplementing Lcarnitine < 4,200 mg/week (60 participants)…” to “Lcarnitine > 4,200 mg/week...”

Reviewer #2: Article written in a clear, objective way and with a robust statistical analysis, performed correctly and adequately for the present study.

Minor corrections:

RESULTS

Characteristics and quality of the studies (page 3)

The total number of participants in the 8 studies is 225 (not 224).

In the phrase "The trials varied in sample size from 9 to 82" patients, the correct one would be "The trials varied in sample size from 9 to 83".

Effect of L-carnitine on dialysis-related hypotension (page 4)

In the sentence "Eight study arms with 224 participants....", the correct would be 225 participants.

In the sentence "In the subgroup analysis [...] while 5 studies supplementing L-carnitine intravenously (47 participants) failed to reveal a difference in incidence of dialysis-related hypotension between these two groups (pooled OR = 0.51, 95% CI [ 0.25-1.06], p = 0.07, I2 = 42.7%)", the correct would be "[...] while 5 studies supplementing L-carnitine intravenously (178 participants)".

In the sentence "Three studies supplementing Lcarnitine > 4,200 mg/week (60 participants)....", the correct sentence would be "Three studies supplementing Lcarnitine < 4,200 mg/week (60 participants)..."

Page 5

In the sentence "Five studies supplementing ≤ 4,200 mg/week (164 participants) showed a non-significant difference....", the correct one would be "Five studies supplementing ≤ 4,200 mg/week (165 participants) showed a non-significant difference...".

In the sentence "Five studies with a duration of treatment at least 12 weeks (175 participants) showed a lower....", the correct sentence would be "Five studies with a duration of treatment at least 12 weeks (176 participants) showed a lower. ...."

Effect of L-carnitine on muscle cramps

In the sentence "Five study arms [...] to control group (pooled OR = 0.22, 95% CI [0.06-0.81], p = 0.02, I2 = 74.7%). (Table 3)", the correct percentage value would be 74.9%.

In table 1, add the meanings of the abbreviations that appear (UK; NA; IV).

In table 3, the values present in the "sample size" column need to be corrected:

Dialysis-related hypotension: 225 and not 224

Route IV: 178 and not 177

Dose ≤ 4200mg/week: 165 and not 164

Duration ≥12 weeks: 176 not 49

Duration < 12 weeks: 45 not 176

6. PLOS authors have the option to publish the peer review history of their article (what does this mean?). If published, this will include your full peer review and any attached files.

Reviewer #1: **Yes: **Sultana Shajahan

Reviewer #2: No

---

## [Author Response · Author response to Decision Letter 0]

14 Jun 2022

Responses to reviewer 1’s comments

In the present systematic review and meta-analysis, the authors investigated the treatment effect of L-carnitine on dialysis-related hypotension and muscle cramps in chronic hemodialysis patients. They included 8 randomized controlled trials with 224 participants, where the mean age of patients ranged from 43.8 to 66.9 years and the duration of follow-up spanned from 6 weeks to 24 weeks. They found a significantly lower odds of dialysis-related hypotension among L-carnitine supplementation group compared to placebo (pooled OR = 0.26, 95% CI [0.10-0.72], p = 0.01, I² = 76.0%), though with high heterogeneity. Further high-quality studies are needed to conclusively prove the effect of L-carnitine on muscle cramps.

The paper is well written, the methodological approach is appropriate and follows PRISMA guidelines, the results are adequately commented upon, and limitations have been correctly examined. The authors have further performed a trial sequential analysis, which helped to conclude the effect of L-carnitine on dialysis-related hypotension. Moreover, subgroup analyses by dose, route and duration of treatment further revealed that oral L-carnitine supplementation at least 12 weeks with a dose above 4,200 mg/week could prevent dialysis-related hypotension.

Response: Thank you for your thoughtful comment.

I have a few minor comments about the manuscript, which I have presented below:

1. Page 5: The authors have stated that “All relevant data are within the manuscript and its Supporting Information files.” Thus, the authors should please amend the data availability statement on Page 4 “No - some restrictions will apply.” PLOS ONE requires all data underlying the findings described fully available.

Response: We amended as suggested. The extracted data file was provided in the supplementary material S3 File.

2. Page 12: Though the authors have found that the lower odds of dialysis-related hypotension among L-carnitine supplementation group compared to placebo was significant, there was high heterogeneity. The authors should please mention this here.

Response: Thank you for your suggestions. We acknowledge the fact that there was high heterogeneity in our study, which led us to apply a random-effects model to analyze the data, incorporating the additional sources of variability between the studies. Despite the fact that we performed subgroup analysis on route, dose and duration of supplementation, the heterogeneity remains. As we mentioned in the discussion, the different definitions in each study for dialysis-related hypotension and muscle cramps may introduce heterogeneity. Moreover, medications such as antihypertensive medications and cardiac function were not specified in each trial which could be important confounders. However, the randomized nature of the data we are synthesizing provides a reasonable amount of confidence that the comparisons performed are not heavily affected by these factors. Nevertheless, we included this lack of information as a limitation in our discussion section. We are also aware that some RCTs had different study designs and dialysis techniques prior to 1990 were different from current practices, which led us to perform a subgroup analysis. The following statements were added in the discussion.

“Second, we acknowledge the heterogeneity in our study was high even after performing subgroup analysis and utilizing a random effect model. The residual confounders such as medications, cardiac function, and different definitions in each study for dialysis-related hypotension and muscle cramps may introduce heterogeneity.”

3. Page 13: In subgroup analysis by dose of L-carnitine supplementation, the following statement requires correction from “Three studies supplementing Lcarnitine < 4,200 mg/week (60 participants)…” to “L-carnitine > 4,200 mg/week...”

Response: Thank you for pointing it out. The following statements were edited in the revised manuscript.

“Three studies supplementing L-carnitine > 4,200 mg/week (60 participants) demonstrated a significant lower odds of dialysis-related hypotension comparing L-carnitine group to control group (pooled OR = 0.03, 95% CI [0.001-0.58], p = 0.02, I2 = 90.0%). Five studies supplementing ≤ 4,200 mg/week (164 participants) showed a non-significant difference in dialysis-related hypotension between treatment and placebo group (pooled OR = 0.78, 95% CI [0.39-1.57], p = 0.48, I2 = 0%). (Table 3)”

Response to reviewer 2’s comments

Article written in a clear, objective way and with a robust statistical analysis, performed correctly and adequately for the present study.

Response: Thank you for your time and effort reviewing this manuscript.

RESULTS

Characteristics and quality of the studies (page 3)

The total number of participants in the 8 studies is 225 (not 224).

In the phrase "The trials varied in sample size from 9 to 82" patients, the correct one would be "The trials varied in sample size from 9 to 83".

Response: Thank you for pointing this out. After careful review of the numbers, we realized the number of participants for the study by Ahmad et al. in Table 1 was 82, not 83. Therefore, the total number of participants in the 8 studies is 224. The trials varied in sample size from 9 to 82. With this opportunity, we went back and checked all the data thoroughly, reassuring that there was no other error.

Effect of L-carnitine on dialysis-related hypotension (page 4)

In the sentence "Eight study arms with 224 participants....", the correct would be 225 participants.

Response: Thank you for pointing this out. We confirmed that our study included 8 study arms with 224 participants.

In the sentence "In the subgroup analysis [...] while 5 studies supplementing L-carnitine intravenously (47 participants) failed to reveal a difference in incidence of dialysis-related hypotension between these two groups (pooled OR = 0.51, 95% CI [ 0.25-1.06], p = 0.07, I2 = 42.7%)", the correct would be "[...] while 5 studies supplementing L-carnitine intravenously (178 participants)".

Response: Thank you for pointing this out. The following sentences were edited as suggested.

“... while 5 studies supplementing L-carnitine intravenously (177 participants) failed to reveal a difference in incidence of dialysis-related hypotension between these two groups (pooled OR = 0.51, 95% CI [0.25-1.06], p = 0.07, I2 = 42.7%). (Table 3)”

In the sentence "Three studies supplementing Lcarnitine > 4,200 mg/week (60 participants)....", the correct sentence would be "Three studies supplementing Lcarnitine < 4,200 mg/week (60 participants)..."

Response: We agreed. The following statements were edited in the revised manuscript.

“Three studies supplementing L-carnitine > 4,200 mg/week (60 participants) demonstrated a significant lower odds of dialysis-related hypotension comparing L-carnitine group to control group (pooled OR = 0.03, 95% CI [0.001-0.58], p = 0.02, I2 = 90.0%). Five studies supplementing ≤ 4,200 mg/week (164 participants) showed a non-significant difference in dialysis-related hypotension between treatment and placebo group (pooled OR = 0.78, 95% CI [0.39-1.57], p = 0.48, I2 = 0%). (Table 3)”

Page 5

In the sentence "Five studies supplementing ≤ 4,200 mg/week (164 participants) showed a non-significant difference....", the correct one would be "Five studies supplementing ≤ 4,200 mg/week (165 participants) showed a non-significant difference...".

In the sentence "Five studies with a duration of treatment at least 12 weeks (175 participants) showed a lower....", the correct sentence would be "Five studies with a duration of treatment at least 12 weeks (176 participants) showed a lower. ...."

Response: As mentioned above, "Five studies supplementing ≤ 4,200 mg/week (164 participants) showed a non-significant difference...." and "Five studies with a duration of treatment at least 12 weeks (175 participants) showed a lower...." were in fact accurate.

Effect of L-carnitine on muscle cramps

In the sentence "Five study arms [...] to control group (pooled OR = 0.22, 95% CI [0.06-0.81], p = 0.02, I2 = 74.7%). (Table 3)", the correct percentage value would be 74.9%.

Response: Thank you for pointing this out. After a careful review of the numbers, we realized that the number in Table 3 was a typo. The I2 for muscle cramps was 74.7%.

In table 1, add the meanings of the abbreviations that appear (UK; NA; IV).

Response: Thank you for your comments. Abbreviations were spelled out below Table 1.

In table 3, the values present in the "sample size" column need to be corrected:

Dialysis-related hypotension: 225 and not 224

Route IV: 178 and not 177

Dose ≤ 4200mg/week: 165 and not 164

Duration ≥12 weeks: 176 not 49

Duration < 12 weeks: 45 not 176

Response: Thank you for pointing out. We double-checked the numbers. With this opportunity, we went back and checked all the data thoroughly, reassuring that there was no other error.

---

## [Editor Report · Decision Letter 1]

28 Jun 2022

The Effect of Levocarnitine Supplementation on Dialysis-related Hypotension: a Systematic Review, Meta-analysis, and Trial Sequential Analysis

PONE-D-22-07106R1

Dear Dr. Chewcharat,

We’re pleased to inform you that your manuscript has been judged scientifically suitable for publication and will be formally accepted for publication once it meets all outstanding technical requirements.

Kind regards,

Mabel Aoun, MD, MPH

Academic Editor

PLOS ONE
---

## [Editor Report · Acceptance letter]

6 Jul 2022

PONE-D-22-07106R1 

The Effect of Levocarnitine Supplementation on Dialysis-related Hypotension: a Systematic Review, Meta-analysis, and Trial Sequential Analysis 

Dear Dr. Chewcharat:

I'm pleased to inform you that your manuscript has been deemed suitable for publication in PLOS ONE. Congratulations! Your manuscript is now with our production department. 

Kind regards, 

on behalf of

Dr. Mabel Aoun 

Academic Editor

PLOS ONE